# The Challenge of a Recall Program from a Community-Based Hepatitis C Screening Campaign: The Effectiveness in HCV Microelimination

**DOI:** 10.3390/microorganisms12071402

**Published:** 2024-07-11

**Authors:** Cheng-Hung Chien, Tien-Shin Chou, Li-Wei Chen, Chih-Lang Lin, Jia-Jang Chang, Ching-Jung Liu, Shuo-Wei Chen, Ching-Chih Hu, Rong-Nan Chien

**Affiliations:** 1Liver Research Unit, Chang Gung Memorial Hospital, Keelung 204, Taiwan; cashhung@cgmh.org.tw (C.-H.C.); f139859@gmail.com (T.-S.C.); leiwei@adm.cgmh.org.tw (L.-W.C.); wn49792000@yahoo.com.tw (C.-L.L.); nonahu@adm.cgmh.org.tw (C.-C.H.); 2Community Medicine Research Center, Chang Gung Memorial Hospital, Keelung 204, Taiwan; 3Division of Gastroenterology, Chang Gung Memorial Hospital, Keelung 204, Taiwan; g15539@adm.cgmh.org.tw (J.-J.C.); cjliuu@adm.cgmh.org.tw (C.-J.L.); poohlouisebaby@yahoo.com.tw (S.-W.C.); 4Liver Research Unit, Linkou Chang Gung Memorial Hospital and University College of Medicine, Taoyuan 333, Taiwan

**Keywords:** hepatitis C virus, linkage to care, hepatitis C microelimination

## Abstract

The optimal strategy for the microelimination of HCV within community settings remains ambiguous. We evaluated the percentage of participants who achieved linkage to care (LTC) following the conclusion of a screening campaign and examined the diverse factors influencing LTC among these individuals. The effectiveness of recall intervention for the non-LTC population and its barriers were analyzed. We initiated an HCV patient recall program to identify HCV participants who might not be treated after the HCV screening campaign. The program staff recalled HCV participants who were lost to follow-up via telephone from March 2019 to June 2019. They were informed of HCV treatment’s importance, efficacy, availability, and safety. Among 185 participants infected with HCV, 109 (58.9%) obtained LTC. Compared with those who had LTC, those without LTC were older, had lower education levels, were less aware of their HCV infection, less frequently lived in urban areas, and had less health insurance. At the end of the recall program, 125 (67.6%) persons had linkage to care. The proportion of LTC increased by 8.7%. In total, 119 persons had an HCV RNA test, and 82 (68.9%) had viremia. Of the 82 patients with viremia, 78 (95.1%) received antiviral therapy, and 76 (97.4%) achieved a sustained virological response. After a community screening campaign, 59% of participants with anti-HCV-positive tests had LTC. The recall program increased this by 9%. However, 32% of HCV participants still could not be linked to care. Outreach care for non-LTC patients is a method worth trying in order to achieve the microelimination of HCV in rural communities.

## 1. Introduction

The World Health Organization (WHO) has a stated goal of eliminating viral hepatitis C virus (HCV) and B virus (HBV) infection as a public health threat by 2030, defined as a 65% reduction in mortality, a 90% reduction in new infections, diagnosing 90% of all hepatitis infections, and treating 80% of the eligible population [1]. In June 2021, the WHO released Interim Guidance for Country Validation of Viral Hepatitis Elimination, including new absolute targets (with an annual incidence of ≤5 per 100,000 and annual mortality of ≤2 per 100,000) [2]. Access to HCV treatment is improving but remains limited. Worldwide, 7% of those diagnosed (1.1 million) started HCV treatment in 2015. Although the availability of highly effective direct-acting anti-viral agent (DAA) therapy for HCV has ushered in a new era in HCV treatment and prevention, simply making DAAs widely available will not mean all people living with hepatitis C have access to these medicines [3]. The estimated prevalence of anti-HCV was 3.3% (1.8–5.5%) in the general population in Taiwan, with several regional disparities. Taiwan has accelerated its efforts to eliminate hepatitis C since 2018 by committing to achieve the WHO’s 2030 goal of treating 80% of eligible patients by 2025 [4]. There are approximately 120,000 chronic hepatitis C (CHC) patients awaiting to be diagnosed in Taiwan. Accelerating the universal screening program for populations aged 45–79 years and resolving the unawareness issue of HCV infection are becoming challenging and critical on the road toward elimination [5].

Considerable research and efforts have focused on the microelimination of HCV among specific populations, such as patients on hemodialysis, individuals coinfected with HIV, people who inject drugs, migrants, and prisoners. There has been generally less emphasis on community-level settings. Community-based screening could provide CHC screening in populations with lower socioeconomic and educational statuses and poor access to healthcare systems [6]. Many questions remain about the best funding mechanisms to ensure program sustainability and the most effective strategies to ensure program reach, linkage to care (LTC), and access to treatment. Understanding and addressing the specific reasons why patients are lost at each stage is critical if public health and clinical care practitioners hope to affect the outcomes of chronic HCV infection. Evan B. Cunningham and colleagues conducted a systematic review and meta-analysis exploring various interventions to enhance the HCV care cascade. Their findings revealed that patient reminders for HCV testing or treatment were associated with a significantly improved uptake of HCV antibody testing [7]. According to our knowledge, studies that assess the challenges of recall programs aimed at increasing linkage to HCV care are lacking.

In this community-based study, we assessed the proportion of participants who obtained LTC since the end of the screening campaign and examined various factors affecting LTC in these participants. A recall program was adopted for the non-LTC population to achieve HCV microelimination in the community. The outcomes of persons eligible for treatment were also evaluated.

## 2. Materials and Methods

### 2.1. HCV Screening Campaign and LTC

The participant recruitment and sample preservation were performed in the Northeastern Taiwan Community Medicine Research Cohort (NTCMRC, ClinicalTrials.gov Identifier: NCT04839796 accessed on 30 June 2024). We implemented a community outreach campaign from August 2013 to December 2017 in the northeastern coastal region of Taiwan, including four districts (Wan-li, Gong-liao, Rul-fan, and An-le). Three districts were rural townships on the northeastern seaboard, and one was urban. Adult participants (≥30 years old) were recruited from the community via public service announcements, talks to community groups, and notices in clinics. A trained team of interviewers administered a standardized questionnaire to all participants. All participants received a demographic survey, a physical examination, and blood tests. Blood tests included complete blood cell and its differentiation count, liver biochemistry, glucose, lipid profile, HBV surface antigen (HBsAg), and HCV antibody tests. All participants were asked if they had ever been infected with HCV. Participants who were HCV antibody-seropositive and had self-reported HCV infection were considered aware of hepatitis C (aHC). Participants who were HCV antibody-seropositive and denied a history of HCV infection were considered unaware of hepatitis C (uaHC). HCV antibody-positive participants were informed of their results via paper mail or in person. Simultaneously, they were recommended to obtain LTC at the Chang Gung Memorial Hospital (CGMH) Keelung branch or other hospitals. The CGMH Keelung branch in Taiwan is a tertiary referral hospital that provides the standard of care for HCV diagnosis and treatment. 

Serum HCV-RNA levels were determined using the Cobas TaqMan HCV Test v. 2.0 (Roche Molecular Diagnostics, Pleasanton, CA, USA) with a lower limit of quantification of 15 IU/mL. The hepatitis C virus genotype was determined using Cobas HCV GT (Roche Molecular Diagnostic). Hepatic fibrosis stages were assessed using transient elastography (FibroScan^®^; Echosens, Hongkong, China). Screening for comorbidities and contraindications to treatment was performed. Patients found to be eligible for treatment were prescribed interferon-based therapy (IBT) and/or DAA according to the national treatment guidelines at the time. 

### 2.2. Recall Program

Since December 2018, we initiated the “recall program” intervention to identify participants who might not have obtained LTC after screening and enhance their access to treatment. Initially, the program staff reviewed the medical records of all HCV antibody-positive participants in CGMH. People who did not visit a hepatologist or were lost to follow-up were contacted via telephone from March 2019 to June 2019 and were asked if they had visited other clinics to manage hepatitis C. The people who had not obtained LTC were informed of HCV treatment’s importance, efficacy, availability, and safety. The program staff provided education and counseling, scheduled hepatologist appointments, accompanied participants to appointments, and provided information about subsidies for medical expenses. The reasons concerning non-LTC and subsequent follow-up results were recorded. The factors affecting LTC and the effectiveness of the recall program were evaluated.

### 2.3. Description of the HCV Treatment in Taiwan

The Bureau of National Health Insurance (NHI) of Taiwan began to reimburse CHC patients for IBT in 2003. A 48- or 24-week regimen of pegylated interferon plus ribavirin combination therapy has been the standard of care. Since January 2017, DAAs have been reimbursed by the Taiwan NHI program for CHC patients with advanced hepatic fibrosis or compensated cirrhosis. On 1 January 2019, the Taiwan NHI authorized the prescription of DAAs to all Taiwanese citizens with CHC. Sustained virologic response (SVR) was defined as undetectable HCV-RNA for ≥three months after DAA treatment completion or six months after IBT, as indicated in the patient’s medical chart [4]. 

### 2.4. Statistical Analysis

Continuous variables are expressed as means ± standard deviation (SD). Statistical comparisons between groups of patients were assessed by performing Student’s *t*-test for continuous variables and Pearson’s chi-square test for categorical variables. McNemar’s test was used to check if the proportion of LTC after the recall program significantly differed from the same proportion before the recall program. Database manipulation and analyses were performed using SPSS version 19 (IBM SPSS Statistics for Windows; IBM Corp., Armonk, NY, USA). A *p*-value of <0.05 indicated statistical significance. 

## 3. Results 

Among the 6161 participants screened for viral hepatitis infection, 185 (3.0%) were seropositive for the HCV antibody (59 men and 126 women; mean age: 65 years). The prevalence of HCV antibody positivity increased with age and 74% were aged > 60 (Figure 1). Of the 185 positive participants, 23 (12.4%) were also seropositive for HBsAg. Thirty-one (16.7%) participants had received anti-HCV treatment before screening. 

### 3.1. Self-Awareness of HCV Infection and LTC

Among the 185 HCV antibody-seropositive participants, 74 (40.0%) were aware of their hepatitis C infection. The age group from 45 to 60 years old had the highest disease awareness (63.3%), and the age group of >75 years old had the lowest disease awareness (11.5%) (Figure 2). Compared with uaHC participants, aHC participants were younger, had higher education levels, had more private health insurance, had greater alcohol consumption, and had more elevated alanine aminotransferase levels (Table 1). Sex, living in an urban area, and a family history of liver disease were not significantly associated with awareness of HCV serostatus. 

After the screening campaign, 109 (58.9%) participants infected with HCV had obtained LTC, including 93 who had regular follow-ups at CGMH and 16 who had regular follow-ups at other hospitals. The remaining 76 (41.1%) participants had not visited a hepatologist or were lost to follow-up after one visit. Compared with those who had LTC, those without LTC were older (67.0 ± 11.6 vs. 63.7 ± 8.6 years old; t(130.7) = 2.216; *p* = 0.035), had lower education levels (X^2^(4, N = 180) = 10.525; *p* = 0.032), were less aware of their HCV infection (X^2^(1, N = 185) = 14.308; 23.7% vs. 51.4%; *p* < 0.001), less frequently lived in urban areas (36.8% vs. 52.3%; X^2^(1, N = 185) = 4.305; *p* = 0.038), and had less health insurance (53.4% vs. 70.2%; X^2^(1, N = 177) = 5.19; *p* = 0.023) (Table 2). 

### 3.2. Effectiveness of the Recall Program

Attempts were made to contact the 76 non-LTC participants in the recall program but 35 (43.4%) could not be contacted because of incorrect telephone numbers or non-responses, 3 (3.9%) died, 16 (19.7%) subsequently started participating in LTC at CGMH, and 22 (28.9%) still did not start LTC despite our advice. After the recall program, 125 (67.6%) participants infected with HCV obtained linkage to care. The proportion of LTC participants significantly increased by 8.7% (X^2^(1, N = 182) = 16; *p* < 0.001). The reasons those 22 non-LTC participants gave were “I need time to consider whether to start treatment” (*n* = 8), “I will seek treatment by myself” (*n* = 7), “I am not willing to be treated because of old age”, “I am too busy”, or “I do not think it would be good for my well-being” (*n* = 5). Two participants did not keep their outpatient appointments.

### 3.3. Clinical Features and Outcomes of the LTC Participants

Among the 125 LTC participants, 119 (95.2%) were tested for HCV-RNA. Of them, 82 (68.9%) were positive for HCV-RNA. Fifty-five participants had undergone Fiboscan^®^ (Echosens). The liver fibrosis stages were F0 = 15 (25.9%), F1 = 13 (22.4%), F2 = 10 (17.2%), F3 = 11 (19.0), and F4 = 9 (15.5%). The genotype distribution was GT1b = 41 (62.1%), GT2 = 24 (36.3%), and indeterminate = 1 (1.5%). In total, 78 (95.1%) eligible patients received anti-viral therapy, including 36 (46.2%) who received IBT, 3 (3.8%) who had IBT failure and were then retreated with DAA, and 39 (50.0%) who received DAA. All these patients completed treatment and showed end-of-treatment virological responses. Seventy-six (97.4%) patients achieved SVR. Of the 82 HCV-RNA-positive patients, 76 (92.7%) were cured (Table 3; Figure 3). 

## 4. Discussion

Our study aimed to evaluate the HCV care continuum among participants after an HCV screening campaign and the effectiveness of a recall program in Northeastern Taiwan communities. We found that 41% of diagnosed HCV participants did not have LTC. The associated factors affecting LTC included older age, living in a rural area, being unaware of their HCV infection, and a lack of health insurance. Our recall program significantly increased the proportion of patients with LTC by 9%. Among those with LTC, most eligible patients (95.1%) received anti-viral therapy, and 97.4% were successfully treated. 

A lack of awareness of HCV infection among affected individuals might be attributed to the asymptomatic nature of CHC until the development of end-stage liver disease. A population-based study surveyed from 2013 through 2016 in the United States highlighted the continued inadequate state of viral hepatitis awareness (<50%) among all individuals and especially among high-risk subpopulations [8]. Another nationwide community outreach screening program in Taiwan found that 44.6% of people were aware of their infection [9]. In this study, aware subjects were younger, had higher education levels, more health insurance, and greater alcohol consumption. We found that people 45 to 60 years old were more likely to be aware of their HCV status than younger or older people. This finding was consistent with data from NHANES [10]. This result reflects that middle-aged adults are more motivated and likely to be screened for the virus. The more significant amount of alcohol consumed by aware subjects may be related to their age. However, this result reveals that most HCV transmission in Taiwan occurs via the iatrogenic route in the early years [11]. Patients > 60 years old accounted for 74% of all patients in our study but their disease awareness was relatively low. As countries that have successfully treated all diagnosed cases run out of patients without increasing the diagnostic rate via screening, their progress toward elimination will substantially slow [12]. Additional community-based screening strategies still need to be designed to identify and treat more infected individuals.

At the population level, the “cascade of care” is a continuum of recommended services for persons living with hepatitis as they enter various stages, from diagnosis to treatment to chronic care and, for hepatitis C, to cure [1]. A nationwide survey in Taiwan conducted by Yu et al. reported that 36.2% of HCV antibody-seropositive subjects were aware of their diagnosis. Among those with disease awareness, 39.6% had accessibility. The antiviral therapy recommendation/acceptance rate was 70.6% [6]. Our study found that 41% of HCV antibody-seropositive participants did not start LTC after learning of their HCV infection. Of those participants, 29% still chose not to start LTC despite our recall program and recommendations. Previous studies have shown that barriers to LTC and treatment include a lack of trust in healthcare providers [13], misconceptions regarding treatment adherence [14], and higher rates of depression and psychiatric illnesses [15]. 

We addressed three significant factors associated with LTC. First, our study indicated that being unaware of their infection was a predictor of non-LTC. Infected individuals’ movement through the viral hepatitis care cascade is, first, contingent on their awareness of their diagnosis, often the step with the largest drop-off in the cascade [16]. According to Zhou’s study, less than half of those aware of HCV infection reported receiving treatment, identifying an additional gap in the care cascade [8]. Studies based on data from the NHANES hepatitis C follow-up survey also indicated that knowledge about having HCV infection was the only independent predictor of receiving treatment [10,17]. As shown in this study, more than half of the older participants were unaware of their HCV infection and tended to be lost to follow-up. A recent study in Georgia showed that implementing a door-to-door testing approach can increase awareness and ultimately improve the linkage to HCV care [18].

Second, we found that older persons with lower education levels were significantly less likely to have LTC. This result might be because those individuals lacked knowledge about HCV infection and did not recognize themselves as being at risk for liver disease progression. Additionally, poor knowledge of HCV treatment, fear of treatment side effects, and the asymptomatic nature of HCV infection were also the main reasons for not considering HCV treatment. Our recall program elucidated how more individuals infected with HCV were successfully mobilized from screening into care and treatment. Engaging with people in discussions about HCV, its treatment, and the potential long-term consequences of HCV-related liver disease if left untreated individually increased the willingness to seek treatment. This engagement led to 20% of people who previously did not have LTC being successfully referred for HCV care. 

Third, people with less health insurance and living in rural areas were less likely to have LTC. There were differences in the sociodemographic characteristics of patients diagnosed with HCV and in the success rate linking patients to outpatient care. Patients who had already seen a doctor or other healthcare professional were more likely to have health insurance and a usual source of medical care than those who had not [10]. Furthermore, inadequate health insurance coverage and limited access to regular healthcare may limit patients’ health-seeking behaviors [19]. Even if case managers scheduled hepatologist appointments and provided follow-up reminder phone calls, some people still did not start LTC due to economic and transportation considerations. A simplified two-step care cascade enabling a test-and-treat model will enhance HCV treatment scale-up [20]. 

Our epidemiological survey showed that approximately 69% of the HCV antibody-seropositive patients were viremic in the community. We speculated that 128 (69%) of the 185 HCV antibody-positive patients should be treated. Accordingly, we estimated that ≥60.9% of the eligible patients were treated and 59.3% were finally cured. The results did not reach the desired WHO goal. There are areas for improvement in the care continuum to achieve the microelimination of HCV in the community. Although the recall program could increase the uptake of LTC, a proportion of patients were still unreachable or refused LTC. In rural areas of Taiwan, a recent pilot study successfully achieved a link-to-care rate of 94.4%. This accomplishment was facilitated by employing multidisciplinary HCV care teams, which included outreach liver specialists providing decentralized on site services [21]. We suggested outreach care would be an effective strategy for HCV treatment provision in the rural community. For residents with limited medical resources, developing point-of-care HCV-RNA diagnostics may facilitate a real “test-and-treat” model of care. Simplification of the therapeutic pathway continues to evolve, as described in the updated clinical guidelines. Mobile outreach care, including motivational education, non-invasive liver disease assessment, and DAA home-delivery services, would provide convenient services to enhance treatment uptake, particularly for socioeconomically disadvantaged older people.

The present study had several limitations. First, a significant proportion (19%) of the HCV antibody-seropositive participants could not be contacted and were lost to follow-up. Therefore, we cannot be sure if they had participated in LTC by being evaluated or treated in other medical clinics and so the percentage of patients participating in LTC and receiving treatment could have been underestimated. Second, the study’s sample size was small; thus, the intervention cannot be generalized to a larger population. The study purpose, outreach testing capacity, and grant funding limited the number of participants who could be enrolled. The other barriers to a rural community study include inconvenient testing facilities and a lack of advocacy and promotion [22,23]. To scale up the screening and treatment uptake, further intervention and cascade of care should integrate local medical resources, including public health systems, primary healthcare, and pharmacies. Third, the cost and effectiveness of the recall program were not analyzed. The research staff managed the program, and we did not record or estimate the cost of conducting recall activities. Further research including cost-effectiveness analysis will be more informative for public policy making. 

## 5. Conclusions

Our study demonstrated that a high treatment uptake and cure rate can be expected in the DAA era granted that the patient has LTC. However, disadvantaged populations still face obstacles to seeking medical treatment, even after knowing that they are infected with HCV. COVID-19 has resulted in the suspension or delay of the HCV elimination program. It is critical to perform large-scale or universal screening of the population and link patients to a point-of-care facility to scale up treatment coverage. Engaging with people to increase awareness of HCV-related liver disease and scheduling specialist appointments via a recall program have been shown to be effective. Outreach care for non-LTC patients is a method worth trying to achieve the microelimination of HCV in rural communities.

## Figures and Tables

**Figure 1 microorganisms-12-01402-f001:**
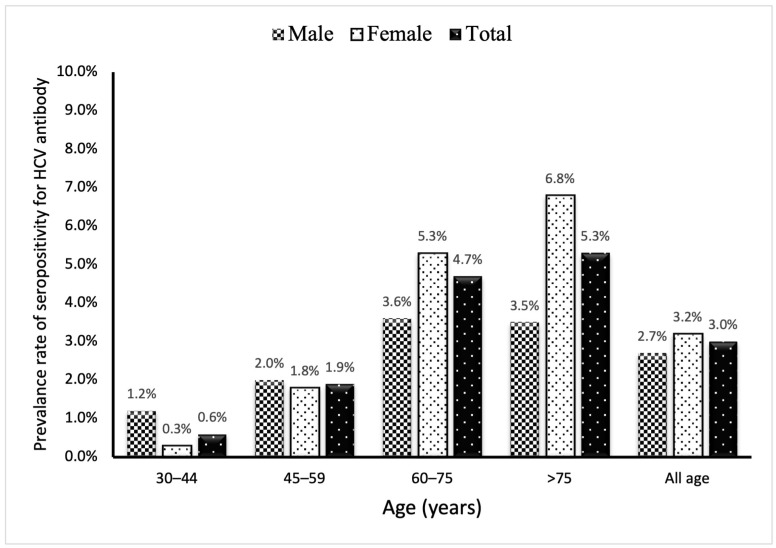
The prevalence rate of seropositivity for the HCV antibody in 6161 participants by age and sex.

**Figure 2 microorganisms-12-01402-f002:**
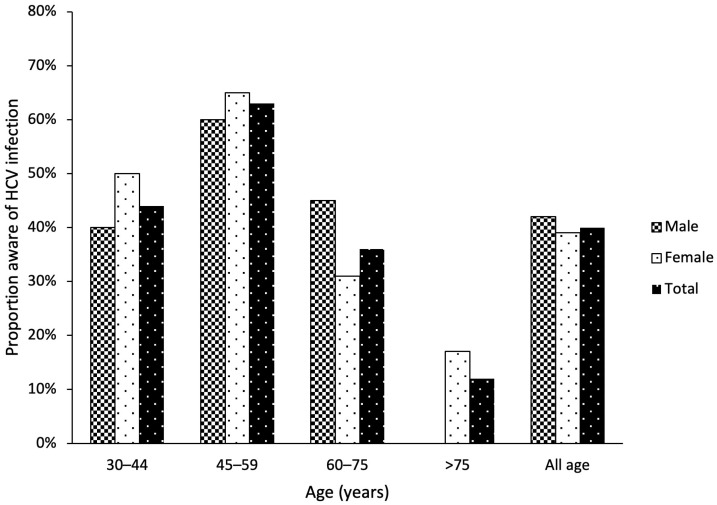
The proportion of 185 participants aware of their HCV infection by age and sex.

**Figure 3 microorganisms-12-01402-f003:**
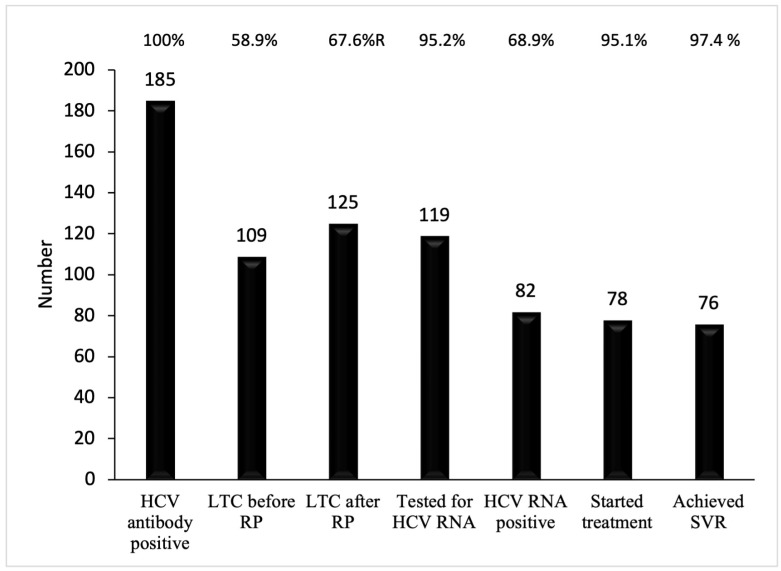
The continuum of viral hepatitis services and the retention cascade after the recall program. The proportion of LTC increased by 8.7% (58.9% to 67.6%) after the recall program. LTC, linkage to care. RP, recall program. SVR, sustained viral response.

**Table 1 microorganisms-12-01402-t001:** Comparison of baseline characteristics and clinical profiles of participants by awareness of HCV.

	Aware of HCV Infection		
	No	Yes	*p*-Value
Number	111	74	
Male (%)	34 (30.6)	25 (33.8)	0.625
Age, years	67.7 ± 10.1	61.0 ± 8.5	<0.001
30–44	5 (4.5)	4 (5.4)	<0.001
45–59	18 (16.2)	31 (41.9)	
60–75	65 (58.6)	36 (48.6)	
>75	23 (20.7)	3 (4.1)	
BMI, kg/m^2^	25.3 ± 3.6	24.3 ± 3.9	0.075
Private health insurance, yes (%)	56 (52.8)	56 (78.9)	<0.001
Education level (%)			
No education completed	29 (26.9)	3 (4.2)	<0.001
Elementary school	39 (36.1)	21 (29.2)	
Middle school	18 (16.7)	21 (29.2)	
High school	12 (11.1)	15 (20.8)	
College and higher	10 (9.3)	12 (16.7)	
Live in urban area (%)	45 (40.5)	40 (54.1)	0.071
Alcohol consumption, yes (%)	16 (14.7)	23 (32.9)	0.04
Smoking, yes (%)	22 (19.8)	19 (26.0)	0.322
Diabetes mellitus (%)	19 (17.1)	14 (18.9)	0.754
Hypertension (%)	44 (39.6)	25 (33.8)	0.42
Hepatitis B carrier (%)	11 (10.0)	10 (13.7)	0.442
Family history of liver disease (%)	12 (10.9)	13 (17.8)	0.183
AST > 34 U/L (%)	25 (22.5)	25 (33.8)	0.091
ALT > 36 U/L (%)	25 (22.5)	27 (36.5)	0.038

**Table 2 microorganisms-12-01402-t002:** Comparison of baseline characteristics and clinical profiles of participants by linkage to care. LTC, linkage to care.

	Linkage to Care		
	No	Yes	*p*-Value
Number	76	109	
Male (%)	26 (34.2)	33 (30.3)	0.572
Age, years	67.0 ± 11.6	63.7 ± 8.6	0.035
BMI, kg/m^2^	24.9 ± 3.3	24.9 ± 4.0	0.982
Live in urban area (%)	28 (36.8)	57 (52.3)	0.038
Education level (%)			
No education completed	21 (28.8)	11 (10.5)	0.032
Elementary school	20 (26.7)	40 (38.1)	
Middle school	13 (17.3)	26 (24.8)	
High school	11 (14.7)	16 (15.2)	
College and higher	10 (13.3)	12 (11.4)	
Insurance, yes (%)	39 (53.4)	73 (70.2)	0.023
Alcohol consumption, yes (%)	15 (19.7)	24 (23.3)	0.568
Smoking, yes (%)	19 (25.0)	22 (20.4)	0.457
Aware of HCV infection	18 (23.7)	56 (51.4)	<0.001
Diabetes mellitus	12 (15.8)	20 (18.7)	0.611
Hypertension	27 (35.5)	40 (37.4)	0.797
Stroke	1 (1.3)	4 (3.7)	0.322
HBsAg(+)	11 (14.5)	12 (11.0)	0.482
Family history of liver disease	13 (17.1)	12 (11.2)	0.253
AST > 34 U/L	16 (21.1)	34 (31.2)	0.127
ALT > 36 U/L	17 (22.4)	35 (34.4)	0.147

**Table 3 microorganisms-12-01402-t003:** Clinical features and outcomes of participants with linkage to care.

Percentage of LTC before recall program	58.9 (109/185)
Percentage of mortality	1.6 (3/185)
Percentage of LTC after recall program	67.6 (125/185)
Percentage of viremia in HCV antibody-positive participants	68.9 (82/119)
Genotype (*n* = 66), % (n)	
Type 1b	62.1 (41)
Type 2	36.3 (24)
Indeterminate	1.5 (1)
Fibrosis stage (*n* = 58), % (n)	
F0	25.9 (15)
F1	22.4 (13)
F2	17.2 (10)
F3	19.0 (11)
F4	15.5 (9)
Percentage of treatment in eligible patients	95.1 (78/82)
Antiviral drugs, % (n)	
Interferon-based	46.2 (36)
Interferon-based then DAA	3.8 (3)
DAA	53.8 (39)
End-of-treatment response, %	100 (78/78)
Sustained viral response, %	97.4 (76/78)
Cured rate in eligible patients, %	92.7 (76/82)

## Data Availability

The original contributions presented in the study are included in the article, further inquiries can be directed to the corresponding author.

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
