# Peer review of "The Challenge of a Recall Program from a Community-Based Hepatitis C Screening Campaign: The Effectiveness in HCV Microelimination"

_microorganisms, 2024, doi:10.3390/microorganisms12071402_

Round 1

Reviewer 1 Report

Comments and Suggestions for Authors

1. Explain correlation between anti-HCV and HCV RNA qualitative (screening) and quantitative (viral load) tests. Since you took only anti-HCV positive samples. Were there any possibilities detect by HCV RNA and missing by anti-HCV?

2. Includes threshold value for viremia and treatment monitoring, antiviral therapy?

Author Response

  1. We conduct large-scale HCV screening in the community. Every resident is tested for anti-HCV for screening, and the seropositive subjects are transferred to the hospital for HCV RNA testing and treatment. This study showed that 68.9 % of anti-HCV seropositive subjects had detected HCV RNA. In patients with chronic hepatitis C, anti-HCV is persistent in serum for life. Using anti-HCV to find people with hepatitis C in community screening should be sufficient.
  2. In the material and methods (2.1 the last paragraph), serum HCV-RNA level was determined by the Cobas TaqMan HCV Test v. 2.0 with a lower limit of quantification of 15 IU/mL. HCV viremia was determined by detectable HCV RNA. Instructions related to treatment are written in material and methods (2.3)

Thank you very much for your kind comments on our manuscript.

Reviewer 2 Report

Comments and Suggestions for Authors

The results in the abstract are described superficially, and it is unclear how significant they are. The introduction is written very superficially. Only single references are indicated. It is not clear what the innovation of this article is and what has been published so far.

In the results, it is unclear where and which statistical test was applied. Results should be reported according to scientific standards, e.g., t(20) = 3.89; p < 0.001. The most important results should be supported by appropriate effect size measures for the applied statistical tests. The p-value alone is definitely insufficient. It is unclear how practically significant the obtained results are.

The conclusions should be shortened and improved based on the obtained effect size measures. At this moment, the conclusions do not follow from the obtained results and are exaggerated.

The references in the article do not include the latest publications, of which there are many. The introduction and discussion should be enriched with the latest publications.

Comments on the Quality of English Language

Minor editing of English language required.

Author Response

  1. We agree that listing relevant past research literature in the background is essential. We incorporate a recent systematic review and meta-analysis study in the introduction section (Cunnungham 2022) to establish the motivation of our study. We believe that assessing each stage of the HCV "cascade of care" is crucial for understanding the effectiveness of interventions and strategies. As a result, we have included the findings from our study in the abstract.
  2. In this study, many of the data are descriptive statistical results, and we have not yet found a better way to analyze our data. Thank you for your statistical advice.
  3. We admit the conclusion should be shortened. We deleted some sentences that can be ignored and modified the article.
  4. We have update and incorporated the four latest publications (WHO 2021, Cunnungham 2022, Butsashvili, 2022, Lo 2023) to enrich the of the introduction and discussion, providing a more comprehensive overview of the topic.

Reviewer 3 Report

Comments and Suggestions for Authors

- WHO Global hepatitis report 2017 should be updated;

- Information about cost and comparison whit similar programmes in other countries should be reported;

- in the age groups 30-44 and 45-59 the number of infected female was higher than that of infected male: this is uncommon for HCV worldwide and should be deeper investigated and discussed;

Comments on the Quality of English Language

the quality of english language including typing errors should be revised.

Author Response

  1. We incorporate a 2021 WHO interim guidance for country validation of viral hepatitis elimination in introduction.
  2. While there has been considerable research and efforts focused on micro-elimination of Hepatitis C virus (HCV) among specific populations such as patients with hemodialysis, HIV coinfected individuals, people who inject drugs, migrants, and prisoners, there is generally less emphasis on community-level projects. According to our knowledge, there is currently a lack of studies that assess the cost-effectiveness of recall programs aimed at increasing linkage to HCV care. The cost of implementing such interventions is likely to differ depending on factors such as disease prevalence, setting, and available resources. Unfortunately, we have not identified any similar studies to provide a basis for comparison. However, a systematic review and meta-analysis conducted by Evan B Cunningham and colleagues explored various interventions designed to enhance the HCV care cascade. Their findings revealed that patient reminders for HCV testing or treatment were associated with a significant improvement in the uptake of HCV antibody testing. We have included this important report in our introduction to support our discussion on the potential effectiveness of recall programs for HCV care.
  1. In the age groups 30-44 and 45-59, the prevalence of HCV infection was found to be higher in males compared to females. These findings are consistent with the majority of studies.

Thank you so much for your thoughtful and helpful feedback on our manuscript.

Round 2

Reviewer 2 Report

Comments and Suggestions for Authors

"In the results, it is unclear where and which statistical test was applied. Results should be reported according to scientific standards, e.g., t(20) = 3.89; p < 0.001. The most important results should be supported by appropriate effect size measures for the applied statistical tests. The p-value alone is definitely insufficient. It is unclear how practically significant the obtained results are." - This type of recommendation was not taken into account.

Comments on the Quality of English Language

Minor editing of English language required.

Author Response

Comment 1: In the results, it is unclear where and which statistical test was applied. Results should be reported according to scientific standards, e.g., t(20) = 3.89; p < 0.001. The most important results should be supported by appropriate effect size measures for the applied statistical tests. The p-value alone is definitely insufficient. It is unclear how practically significant the obtained results are

Response 1: Thank you for point this out. We updated the statistical results in page 6 line 158-162 and page 7 line 171. 

Comment 2: Minor editing of English language required.

Response 2: Agree. We used MDPI's english editing service and carefully checked all changes in the revised manuscript. 

Reviewer 3 Report

Comments and Suggestions for Authors

Dear author, not all the criticisms were resolved. In the text and Fig.1 of the revised manuscript still remain higher the percentage of HCV infected female compared to male: this didn't correspond to what you reported in the rebuttal letter where the opposite was written. The manuscript was not improved and the poor novelty remains the main unresolved criticism.

Comments on the Quality of English Language

English typing errors are still present and generally the quality of the language should be improved: repetitions of sentences, etc.

Author Response

Comment 1 : Not all the criticisms were resolved. In the text and Fig.1 of the revised manuscript still remain higher the percentage of HCV infected female compared to male: this didn't correspond to what you reported in the rebuttal letter where the opposite was written.

Response 1: We carefully checked our report again. The data indicates that in the age group of 30-44, the prevalence of HCV infection is higher in males compared to females, with rates of 1.2% and 0.3% respectively. Similarly, in the age group of 45-59, the prevalence of HCV infection is higher in males compared to females, with rates of 2.0% and 1.8% respectively. We have reordered the figure 1 to show the prevalence of HCV infection among different age groups and genders.

Comment 2: The manuscript was not improved and the poor novelty remains the main unresolved criticism.

Response: Thank you for your comment and feedback. We acknowledge the importance of enhancing the novelty of this study. To address this, we have included two additional paragraphs in the introduction, specifically on page 2, lines 51-54 and lines 64-65. These paragraphs aim to emphasize the uniqueness of our research and contribute to the existing scientific literature. We appreciate your suggestions for improvement.

Comment 3: English typing errors are still present and generally the quality of the language should be improved: repetitions of sentences, etc.

Response: Agree. We used MDPI's English editing service and carefully checked all changes in the revised manuscript.